# Understanding the psychological experiences of loneliness in later life: qualitative protocol to inform technology development

Jessica Rees ![ORCID],[1] Wei Liu,[2] Sebastien Ourselin,[3] Yu Shi,[4] Freya Probst,[2] Michela Antonelli,[3] Anthea Tinker,[1] Faith Matcham[5]

¹Department of Global Health and Social Medicine, King's College London, London, UK
²Department of Engineering, King's College London, London, UK
³School of Biomedical Engineering & Imaging Sciences, King's College London, London, UK
⁴School of Design, University of Leeds, Leeds, UK
⁵School of Psychology, University of Sussex, Falmer, UK

**Correspondence to**
Dr Wei Liu; wei.liu@kcl.ac.uk

## ABSTRACT

**Objectives** Loneliness is a public health issue impacting the health and well-being of older adults. This protocol focuses on understanding the psychological experiences of loneliness in later life to inform technology development as part of the 'Design for health ageing: a smart system to detect loneliness in older people' (DELONELINESS) study.
**Methods and analysis** Data will be collected from semi-structured interviews with up to 60 people over the age of 65 on their experiences of loneliness and preferences for sensor-based technologies. The interviews will be audio-recorded, transcribed and analysed using a thematic codebook approach on NVivo software.
**Ethics and dissemination** This study has received ethical approval by Research Ethics Committee's at King's College London (reference number: LRS/DP-21/22-33376) and the University of Sussex (reference number: ER/JH878/1). All participants will be required to provide informed consent. Results will be used to inform technology development within the DELONELINESS study and will be disseminated in peer-reviewed publications and conferences.

## INTRODUCTION

### Loneliness: definition, impact and prevalence

Loneliness has been defined as 'an unwelcome feeling of lack or loss of companionship'.[1] This is based on one theoretical framework for loneliness, where a person experiences a 'mismatch' (or cognitive discrepancy) between actual and desired levels of social contact or quality of relationships.[2] The subjective nature of loneliness means it is experienced and felt differently by each individual. This is distinct from the concept of social isolation, which quantifies the number of connections in a person's social network. Social isolation is most closely aligned with social loneliness, defined as an objective number of social contacts and connections.[3] Emotional loneliness, which can also be relationship-specific, refers to the lack or loss of meaningful, good quality relationships.[4] More recent conceptualisations

---

**STRENGTHS AND LIMITATIONS OF THIS STUDY**

⇒ A strength of this study is the large sample (n=60) which we will use to stratify experiences of loneliness in later life which correlate with technology preferences.
⇒ We have integrated patient and public involvement into the design and conduct of our study.
⇒ The data collection methods presented in this protocol are inclusive to facilitate the participation of older adults based on technology preferences.
⇒ One limitation of this study is that conceptualisations of loneliness may exclude people from taking part in interviews.

---

have also described existential loneliness as a sense of separateness from others.[5]

Loneliness is a rising public health issue[6] with detrimental impacts on an individual's health.[5] Older adults experience both physical and psychological changes as a result of loneliness.[7] There is consistent evidence of the impact of loneliness on important health outcomes such incidence of cardiovascular disease,[8] risk of dementia,[9] and multimorbidity[10] leading to increased mortality.[11] Loneliness is an unpleasant experience[2] associated with negative emotions such as worry and sadness.[5] As such, the bidirectional nature of loneliness and mental health is well known. Specifically conditions such as depression and anxiety, and symptoms such as sleep difficulties, eating or substance use disorders and suicidal ideation are more frequently reported in people experiencing loneliness.[12]

Loneliness is a common experience in later life, with one in four people over the age of 65 experiencing moderate loneliness in high-income countries.[13] In the UK, this translates to 1.4 million older adults experiencing loneliness often.[14] Those at greater risk of feeling lonely include widowed older homeowners who live alone with a long-term health

condition.[15] The prevalence of loneliness in older adults reportedly increased from 26% to 32% after the first 3 months of the COVID-19 pandemic[16] as social distancing restrictions changed social networks and support.[17]

## Experiences of loneliness

Qualitative research provides useful insights into the nuanced, complex and multi-faceted nature of loneliness. A meta-synthesis of how older adults experience and manage loneliness described how interpersonal relationships are linked with negative emotions such as helplessness, sadness, grief, disappointment.[18] Loss is an important aspect of loneliness for older adults, a factor which has been outlined in the Social Relationship Expectation framework.[19] Loneliness has also been described as an 'unspoken and trivialised' experience due to stigma which is enhanced by public discourse around ageing and health.[20]

Coping strategies for loneliness employed by older adults range from prevention and action to acceptance and endurance.[21] These dynamic approaches depended on whether an individual preferred coping alone or with others. Those who addressed loneliness with others were more likely to engage with services and social activities. This has implications for services to identify and provide support for individuals experiencing loneliness yet prefer to cope alone. On an individual levels, cognitive strategies (such as acceptance) can facilitate management of negative feelings associated with loneliness.[22] Emotional regulation strategies for loneliness have parallels with psychological distress, although the ability to change one's thinking about a situation (ie, cognitive reappraisal) has been found to be low in people experiencing loneliness.[23]

## Risk factors for loneliness

There is a vast literature on the risk factors or predictors of loneliness for older adults. Categories include socio-demographic factors (eg, age, gender, income), psychological attributes (eg, neuroticism, self-efficacy), social resources (eg, social contacts, marital status) and health attributes (eg, health/functional status).[24 25] Researchers have sought to use such factors to develop profiles of individuals most at risk of loneliness.[26 27] For example, older adults who are not married/widowed, those with limited social contact/support and have poor self-report health.

Psychological factors relating to behaviours, feelings, thoughts and attitudes[28] are also relevant to the categorisation of loneliness. These include an individual's attribution style (ie, internal or external explanations for life events), coping style (ie, problem or emotion focused), personality characteristics (ie, neuroticism, early life experiences) and self-esteem/efficacy (ie, belief in own ability to succeed in social situations).[12] Lower loneliness levels have been found in people with extroversion, positive mental well-being, informal social contacts and where spouse is a close confidant.[29] These studies are important to understand how we might measure loneliness and to increase our understanding of interacting effects.[30]

## Technology and loneliness

To develop effective interventions, it is important for future research to consider context-specific predictors for the dimensions of loneliness which map onto interacting explanations for behaviour.[31] There is growing interest in the use of technology to detect loneliness. This is based on the principle of early intervention to prevent further physical or mental decline. Previous research has measured daily life patterns to detect loneliness using either smart-based methods (eg, ambient sensors to measure motion and touch) or smartphone and wearable-based sensors.[32] These include mobile phone use, time spent in/out of home, sleep habits, mobility patterns, proximity sensing and conversation activity.

A recent systematic review identified seven studies which used sensor-based technologies (smart-homes) to detect or predict feelings of loneliness in older adults.[33] Digital phenotypes of behaviour or daily activities can be monitored, then algorithms can be used to infer loneliness by deriving behaviour patterns from data. Such approaches have been described as 'promising research path for overcoming loneliness' [33 (p.10)]; however, important ethical and privacy issues still remain to be addressed in future work.

To develop detection methods using machine learning techniques, previous research used a detailed understanding of the psycho-physiological signs and symptoms of stress.[34] By further exploring the experiences of loneliness in later life, we aim to develop a context-specific understanding of the psychosocial parameters of loneliness which will inform future technology development.

## This protocol

This qualitative study explores the psychological experiences of loneliness in older adults and associated preferences for technology useability and engagement. Our findings will inform the development of a smart monitoring and communication system with multifunctional electronics built into textiles used as wearables and home furniture to measure loneliness levels in older people. Critical to the success of developing a novel smart monitoring system is involving end-users in the development and design of the system.

To gather feedback about the suitability of a smart system to detect loneliness, we aim to develop a holistic perspective of the psychological experiences of loneliness in later life, and how this relates to willingness to interact with sensor-based technologies. Our specific objectives include:

► To explore older people's experience of loneliness across a number of psychological and social parameters and behaviours.
► To describe the context and circumstance in which a smart system might be most useful and useable in an elderly population.
► To identify the most meaningful way to relay data collected by the smart system back to individuals, their family or health and social care providers.

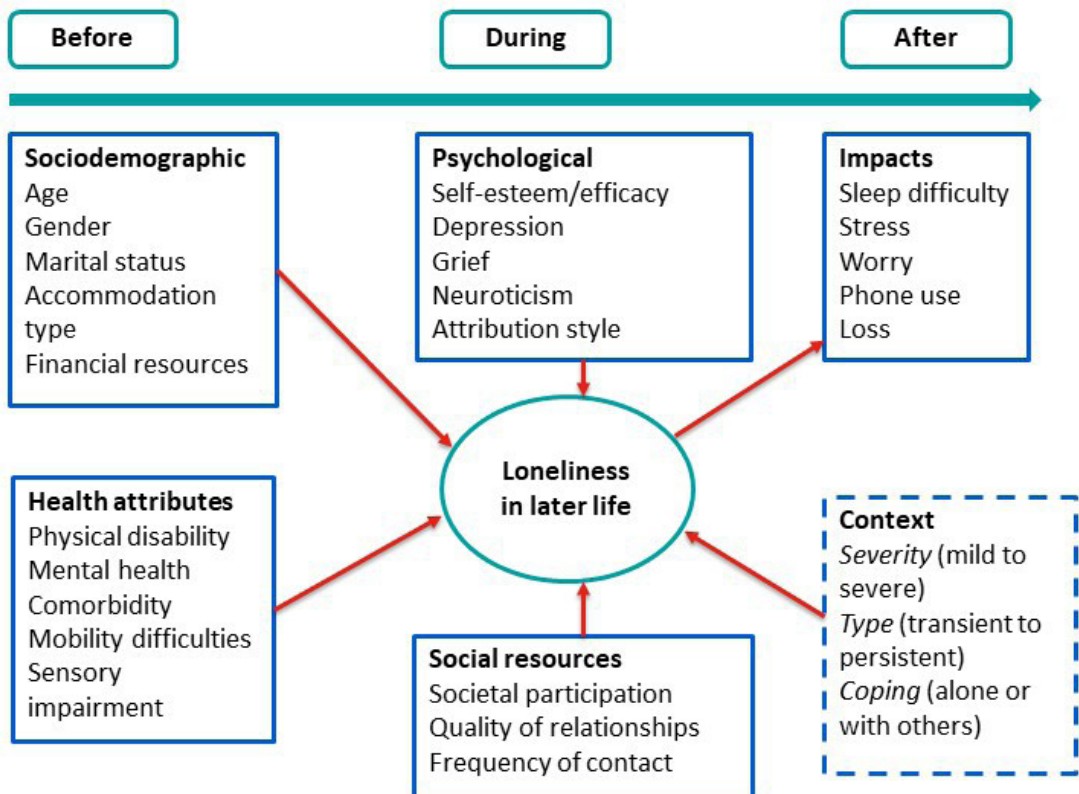

**Figure 1** An evidence-based conceptual model of loneliness.

In figure 1, we outline the Design for health ageing: a smart system to detect loneliness in older people (DELONELINESS) study conceptualisation of loneliness and model for the psychological aspects of loneliness in later life. We aim to analyse these factors to understand what works for who in which circumstances relating to (A) The experiences of loneliness in later life and (B) The preferences of older adults for sensor-based technologies to detect and predict feelings of loneliness.

## METHODS AND ANALYSIS

Due to the subjective nature of loneliness,[5] we will use a qualitative approach to explore the psychosocial parameters of loneliness which are associated with technology preferences in later life. Data will be collected using semi-structured interviews with people over the age of 65. To describe and stratify a range of experiences, we will interview up to 60 individuals with lived experience of loneliness in later life. The discussions will be semi-structured to encourage participants to explore ideas and perspectives, while remaining focused on the research aims. Analysis of interview data will form the basis of recommendations to develop a smart system to monitor loneliness. Data collection for this study began in October 2022 and will end in August 2023.

## Recruitment

Participants will be eligible to take part in interviews if they are over the age of 65, self-identify as having experienced loneliness since reaching the age of 65, are able to give informed consent (ie, no cognitive impairment or dementia), and speak English at a level sufficient for participation. We will purposively sample based on age, gender, accommodation type (own home, rented, sheltered accommodation) and level of digital ability. The latter will be assessed indirectly based on participants communication preferences (ie, telephone verses email).

Study information will be sent to participants from two previous research projects who have provided consent to be contacted for future research. First, participants from the Remote Assessment of Disease and Relapse-Major Depressive Disorder programme which aims to examine whether wearable technology can improve symptom measurement in long-term conditions.[35] Second, participants from PROTECT which is a research project investigating precursors to dementia in an ageing population. We will also share study information via national newsletters sent by email from the Housing Learning and Improvement Network, who provide specialist housing and care solutions for older adults. Finally, we will share online advertisements on research participation websites such as MQ Mental Health. For all recruitment sources, potential participants will be provided with the contact information of the research team and invited to contact us if they are interested in finding out more information. Written informed consent will be obtained from all participants prior to data collection. We will be explicit about the focus on loneliness in all recruitment material.[36]

## Data collection

### Questionnaire data

We will ask sociodemographic information such as age, gender, marital status, ethnicity, accommodation type and educational qualifications. Using a general data protection regulation (GDPR) compliant survey software system (Qualtrics), the researcher will enter verbal responses from participants. To develop a broad understanding of factors related to loneliness and technology preferences, we will record details of participants social environment (number of people seen in a typical week) and medical history (history of depression, physical activity, average sleep, details of any illnesses or disabilities).

We followed specific guidance on selecting scales to measure loneliness in later life.[37] To develop our understanding on the conceptualisations of loneliness, participants will verbally provide answers to two recommended loneliness measures. Both measures will be used to validate participants self-identification of loneliness experiences and indicate level of loneliness severity at the time of interview. In addition, we will ask about mental health and service use. Details of items and scoring are provided below.

### The De Jong Gierveld Loneliness scale

Designed for use in older adult populations,[3] we will use the 11-item scale which is a valid and reliable measure of social loneliness (5 items) and emotional loneliness (6 items). Positively worded items (all of the time, often, some of the time) are counted for emotional loneliness statements, while negatively worded items (none of the time, rarely, some of the time) are counted for social loneliness statements. The total loneliness score is calculated by taking the sum of the emotional loneliness score and the social loneliness score, which can be categorised into four levels: not lonely (score 0, 1 or 2), moderate lonely (scores 3–8), severe lonely (score 9 or 10) and very severe lonely (score 11).[4]

### The Office for National Statistics recommended measures

Following recommendations of the Office for National Statistics, we will use three questions from the University of California, Los Angeles three-item loneliness scale[38] relating to companionship, feeling left out and feeling isolated from others. Response options include 'hardly ever or ever', 'some of the time' and 'often' scored as 1, 2 and 3, respectively. Scores can be added together and grouped into two categories[39]: lonely (scores 6–9) and not lonely (scores 3–5). We then ask a direct measure of loneliness with response options of 'often/always', 'some of the time', 'occasionally', 'never'.[40]

### The Patient Health Questionnaire Anxiety and Depression scale

We will use the four-item scale[41] of this reliable and valid consisting of two core depression items and two core anxiety items. Participant report how symptoms have affected them over the past 2 weeks with response options of 'not at all', 'several days', 'more than half the days' and 'nearly every day', scored as 0, 1, 2 and 3, respectively. Total score is determined by adding together the scores of each of the four items. Scores are rated as normal (0–2), mild,[3–5] moderate[6–8] and severe.[9–12] Total score ≥3 for first two questions suggests anxiety. Total score ≥3 for last two questions suggests depression.[42]

### The Modified Client Services Receipt Inventory for healthcare service use questionaire

The Client Service Receipt inventory is a widely used health resource measurement tool.[43] We will use the modified version to record consultations with healthcare practitioners and hospital admissions, and medication taken (within 3 months). We will also record help received as consequence of health problems (childcare, personal care and help in/outside home).[44]

### Qualitative interview

To increase accessibility for those who cannot travel or do not have access to the internet, interviews will be held either in-person (at participants homes/in university offices) or remotely (via Microsoft Teams video call or telephone call). The format of interviews will depend on participant need and preference. Interviews will last no longer than 120 min, with opportunities for comfort breaks provided. Participants will receive a £30 voucher for taking part in the interviews, to represent the time dedicated to providing data.

The interview guide (see online supplemental material) will include specific questions on participants personal experience of loneliness including definition, correlates, precursors and support received. We developed the topic guide with a multidisciplinary team with expertise in psychology, gerontology, product design, smart composite materials and artificial intelligence. In line with recent guidelines on interviewing older adults about loneliness,[36] we will begin the interview using a third-person approach by asking participants how they define the term 'loneliness'. We will then move onto exploring participants own personal experiences. To co-construct narratives of loneliness, we will prompt participants about their responses to the loneliness items and asked to reflect and expand on their experiences.

The second part of the interview will explore the role of technology in measuring loneliness. To inform product design, we will ask for descriptions of participants living environment, daily routine, existing use of technologies for health and preferences for wearables or sensors in furniture or clothing. The final section will focus on the use of data, where we ask participants for their preferences of how data will be feedback to older adults, circumstances where data is shared with family members and/or health and social care professionals, and thoughts/feelings about being alerted to risk of becoming loneliness.

The interviewer will always end the interview of a positive note.[36] If a sensitive topic is disclosed shortly before the end of the interview, the researcher will check in with the participant and go back to a positive topic previously

discussed. If the researcher has safety concerns, they will make it clear whether the participant should expect a follow-up email or telephone call. Interviews will be audiorecorded and transcribed verbatim by an external company.

## Data analysis

Analysis will be informed by the theory gleaning stage of realist methodology which aims to understand 'what works for whom in which contexts'.[45] Explanatory IF-THEN statements will be developed and tested in an iterative process. For example, IF participants experienced severe and persistent loneliness THEN usefulness of sensor was low due to perceived inability to change personal circumstances. Previous research has used realist evaluation to develop a personalised approach to intervention development in older adults due to the variety in cause and consequence of loneliness.[46] Realist configurations (context-outcome-mechanisms statements) aim to develop key clusters of properties or attributes which underpin cases.[47] We will use this to develop a context-specific understanding of the psychosocial parameters of loneliness which relate to the barriers and facilitators for sensor-based technology uptake.

We will use a codebook approach to qualitative analysis which combines reflexive thematic analysis with structured early theme development.[48] We will use our model of loneliness to generate initial codes based on typologies of loneliness. We will then analyse interview data coding relevant data into these topic summaries. Throughout we will use inductive approaches to review themes and define and name themes which were not present in the codebook. The process of analysis will be iterative by comparing knowledge from the literature with interviews. We will use NVivo software to facilitate the organisation of codes during analysis. For example, to categorise participants based on severity of loneliness (moderate, severe, very severe).

Analysis will be undertaken by the lead author (JR). The process of familiarisation will begin at begin at the point of interviews with the checking of transcripts. Members of the research team will be consulted in regular biweekly meetings throughout data collection and analysis to discuss developing findings. The use of memo notes will be implemented throughout to record decisions on theme development to increase transparency and share findings with wider team.

## Public and patient involvement

The DELONELINESS project seeks consultation from a public and patient involvement group, which includes over 70 adults with an interest in mental health research and commitment to improve inclusivity by advising on race and ethnicity. In October 2022, four members of the group provided feedback on the recruitment methods outlined in this protocol. The group will be consulted throughout the project, specifically to provide feedback on preliminary findings in July 2023 and strategies for dissemination in October 2023.

## ETHICS AND DISSEMINATION
### Ethical considerations

This study has been approved by Research Ethics Committees at King's College London (reference number: LRS/DP-21/22-33376) and the University of Sussex (reference number: ER/JH878/1). The researcher conducting interviews is skilled in qualitative methods relating to sensitive topics in health and ageing and will adopt an ethical positioning of engagement and mutual respect.[36] Regular clinical supervision with a qualified health psychologist will take place to support the interviewer with emotional aspects of conversations.

Time will be taken at the beginning of each interview to establish a rapport with participants and develop an understand of their own personal definition of loneliness. A risk-protocol is in place for if participants were to elicit an emotional response during interviews. With appropriate consent, the risk protocol includes recording participants general practitioner details. In line with GDPR and data protection requirements, the study has data handling and deidentification processes in place to maintain the confidentiality of data will be explained to participants involved in the interviews.

### Dissemination

The findings will initially be disseminated within the study team via internal reports and meetings attending by all co-authors. The aim will be to share preliminary findings of understandings of how people with varied experienced of loneliness in later life engage with technology including the identification of barriers and facilitators to technology uptake. For example, findings related to participants preferences for placement of sensors will be feedback to the product design workstream, and findings related to signs and symptoms of loneliness will be feedback to the smart textile sensor workstream. For academic dissemination, findings will also be published in peer-reviewed journals and shared at conferences. For public dissemination, we will share findings at meetings and events including webinars run by project partners Housing Learning Improvement Network, and through blogs published via the Campaign to End Loneliness.

**Correction notice** This article has been corrected since it was published online. The corresponding author has been changed from "Jessica Rees" to "Wei Liu".

**Acknowledgements** This research was reviewed by a diverse multicultural Patient and Public Involvement group who have been specially trained to advise on research proposals and documentation through the Race, Ethnicity And Diversity group (READ): a free, confidential service in England provided by the National Institute for Health Research Maudsley Biomedical Research Centre via King's College London and South London and Maudsley NHS Foundation Trust.

**Contributors** The study concept and design was conceived by WL, SO, YS, AT and FM. FP and MA assisted in refining the study methodology. JR, AT and FM are responsible for data collection. Analysis will be undertaken by JR with support from

WL, SO, YS, AT, FM, FP and MA. JR prepared the first draft of the manuscript. All authors critically revised the manuscript and approved the submitted version.

**Funding** This work is supported by the UK Engineering and Physical Sciences Research Council (ESPRC) and the National Institute of Health and Care Research (NIHR) (grant number: EP/W031434/1; EP/W031442/1).

**Competing interests** None declared.

**Patient and public involvement** Patients and/or the public were involved in the design, or conduct, or reporting, or dissemination plans of this research. Refer to the Methods section for further details.

**Patient consent for publication** Not applicable.

**Provenance and peer review** Not commissioned; externally peer reviewed.

**ORCID iD**
Jessica Rees http://orcid.org/0000-0002-9471-2134

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
