## [Reviewer comments · BMJ Open]

ARTICLE DETAILS

TITLE (PROVISIONAL)	Understanding the Psychological Experiences of Loneliness in Later Life: Qualitative Protocol to Inform Technology Development
AUTHORS	Rees, Jessica; Liu, Wei; Ourselin, Sebastien; Shi, Yu; Probst, Freya; Antonelli, Michela; Tinker, Anthea; Matcham, Faith

VERSION 1 – REVIEW

REVIEWER	Ibsen, Tanja University of Oslo, Aging and Helath
REVIEW RETURNED	21-Mar-2023

GENERAL COMMENTS	Thank you for letting me review this protocol of a very interesting and important study. The methods and analysis are suitable to answer the aim of this study. I consider that the protocol is in line with the BMJ open's requirements. In reference 17, the year and journal is wrong and must be changed before publication.
---

REVIEWER	Cioffi, Andrea University of Foggia
REVIEW RETURNED	05-Apr-2023

GENERAL COMMENTS	This study protocol is inherent in a particularly important and under-estimated theme. I believe that the protocol drawn up by the authors is rigorous and can provide useful information by expanding knowledge in this area.
---

REVIEWER	Leavey, Gerard University of Ulster, Psychology
REVIEW RETURNED	10-Apr-2023

GENERAL COMMENTS	this is a fairly straightforward protocol description. I am however, unclear why the qualitative student requires supporting with measures of loneliness etc. I understood that recruitment was based on self-identification of loneliness - is there an intention to validate this? Their use is not explained within the Analysis section. Also, I think that the participants' views about technology and how these will be incorporated into the main study could be better explained.
---

VERSION 1 – AUTHOR RESPONSE

Minor revisions

Thank you for reviewing our contribution to BMJ Open and providing us the opportunity to revise it.

Please revise the ‘Strengths and limitations of this study’ section of your manuscript (after the abstract). This section should contain up to five short bullet points, no longer than one sentence each, that relate specifically to the methods. The novelty, aims, results or expected impact of the study should not be summarised here.

Thank you for your comment. We have updated the strengths and limitation section to only focus on methods (pg 1):

- “A strength of this study is the large sample (n=60) which we will use to stratify experiences of loneliness in later life which correlate with technology preferences.
- We have integrated patient and public involvement into the design and conduct of our study.
- The data collection methods presented in this protocol are inclusive to facilitate the participation of older adults based on technology preferences.
- One limitation of this study is that conceptualisations of loneliness may exclude people from taking part in interviews.”

Please include a copy of the interview guide as a supplementary file

A copy of the interview guide has been provided to include as a supplementary file. Signposting to supplementary material for the interview guide has been added on page 7:

“Both measures will be used to validate participants self-identification of loneliness experiences and indicate level of loneliness severity at the time of interview.”

Please embed the following statements to your main document just before your reference list. Contributorship statement, Competing interests, Funding.

These sections have been moved to appear before the reference list as requested.

In reference 17, the year and journal is wrong and must be changed before publication.

The date (2020) and journal (Gerontologist) of reference 17 has been updated.

I am however, unclear why the qualitative student requires supporting with measures of loneliness etc.

Thank you for your comment. In addition to completing the measures specified on pages 6 and 7, the post-doctoral researcher will be completing in-depth interviews with older adults about their experiences of loneliness and preferences of technologies. The section of the interview related to loneliness can be emotional with topics such as grief, suicide and addiction discussed. Therefore, regular clinical supervision is in place to support the researcher as highlighted on page 9:

“Regular clinical supervision with a qualified health psychologist will take place to support the interviewer with emotional aspects of conversations.”

I understood that recruitment was based on self-identification of loneliness - is there an intention to validate this? Their use is not explained within the Analysis section.

Thank you for your suggestion. We have provided further detail on page 6 7 related to the justification of using two recommended loneliness measures (i.e. validated measures to confirm participants self-identification of loneliness):

“Both measures will be used to validate participants self-identification of loneliness experiences and indicate level of loneliness severity at the time of interview.”

In the analysis section, on page 8 we have provided further detail on how severity of loneliness will be incorporated into the analysis:

“For example, to categorise participants based on severity of loneliness (moderate, severe, very severe).”

I think that the participants' views about technology and how these will be incorporated into the main study could be better explained.

We have added further information in the dissemination section on page 9 about how participant views about technology will be incorporated into the DELONELINESS study:

“For example, findings related to participants preferences for placement of sensors will be feedback to the product design workstream, and findings related to signs and symptoms of loneliness will be feedback to the smart textile sensor workstream.”